# Development of a Polymeric Membrane Impregnated with Poly-Lactic Acid (PLA) Nanoparticles Loaded with Red Propolis (RP)

**DOI:** 10.3390/molecules27206959

**Published:** 2022-10-17

**Authors:** Valdemir da Costa Silva, Ticiano G. do Nascimento, Naianny L. O. N. Mergulhão, Johnnatan D. Freitas, Ilza Fernanda B. Duarte, Laisa Carolina G. de Bulhões, Camila B. Dornelas, João Xavier de Araújo, Jucenir dos Santos, Anielle C. A. Silva, Irinaldo D. Basílio, Marilia O. F. Goulart

**Affiliations:** 1Institute of Pharmaceutical Sciences, Federal University of Alagoas (UFAL), Maceio 57072-970, AL, Brazil; 2Institute of Chemistry and Biotechnology, Federal University of Alagoas, Maceio 57072-970, AL, Brazil; 3Department of Chemistry, Federal Institute of Education, Science and Technology, Alagoas, Maceio 57035-660, AL, Brazil; 4Department of Food Technology, Federal University of Viçosa (UFV), Viçosa 36570-900, MG, Brazil; 5Physics Institute, Federal University of Alagoas (UFAL), Maceio 57072-970, AL, Brazil

**Keywords:** polymeric membranes, biopolymers, phenolic compounds, antioxidant activity, propolis

## Abstract

The main objectives of this study were to develop and characterize hydrophilic polymeric membranes impregnated with poly-lactic acid (PLA) nanoparticles (NPs) combined with red propolis (RP). Ultrasonic-assisted extraction was used to obtain 30% (*w*/*v*) red propolis hydroalcoholic extract (RPE). The NPs (75,000 g mol^−1^) alone and incorporated with RP (NPRP) were obtained using the solvent emulsification and diffusion technique. Biopolymeric hydrogel membranes (MNPRP) were obtained using carboxymethylcellulose (CMC) and NPRP. Their characterization was performed using thermal analysis, Fourier transform infrared (FTIR), total phenols (TPC) and flavonoids contents (TFC), and antioxidant activity through the radical scavenging assay with 2,2-diphenyl-1-picrylhydrazyl radical (DPPH) and Ferric reducing antioxidant power (FRAP). The identification and quantification of significant RP markers were performed through UPLC-DAD. The NPs were evaluated for particle size, polydispersity index, and zeta potential. The TPC for RPE, NPRP, and MNPRP was 240.3 ± 3.4, 191.7 ± 0.3, and 183.4 ± 2.1 mg EGA g^−1^, while for TFC, the value was 37.8 ± 0.9, 35 ± 3.9, and 26.8 ± 1.9 mg EQ g^−1^, respectively. Relevant antioxidant activity was also observed by FRAP, with 1400.2 (RPE), 1294.2 (NPRP), and 696.2 µmol Fe2+ g^−1^ (MNPRP). The primary markers of RP were liquiritigenin, isoliquiritigenin, and formononetin. The particle sizes were 194.1 (NPs) and 361.2 nm (NPRP), with an encapsulation efficiency of 85.4%. Thermal analysis revealed high thermal stability for the PLA, nanoparticles, and membranes. The DSC revealed no interaction between the components. FTIR allowed for characterizing the RPE encapsulation in NPRP and CMC for the MNPRP. The membrane loaded with NPRP, fully characterized, has antioxidant capacity and may have application in the treatment of skin wounds.

## 1. Introduction

In the last twenty years, the development of new biomaterials has been marked by constant growth, and several research segments corroborate the availability of new biomaterials. Polymeric biomaterials of natural or synthetic origin are widely used in the development of several products. Among the products that stand out are those that have biomedical, pharmaceutical, dental, and industrial applications as well as uses in the food area, the production of biofilms, and cosmetology [1,2].

Polymers of natural origin have aroused intense technological interest in the development of several bioproducts applied to promote wound healing, such as carboxymethylcellulose, derived from cellulose, which is a low-cost and highly abundant natural polymer. Its main feature is its high liquid absorption capacity. This characteristic is important for the manufacture of biomaterials to accelerate the wound healing process [3,4]. Biodegradable and bioabsorbable synthetic polymeric materials have been extensively investigated, and they are promising in the development of biomedical devices for application at the heart of tissue recovery [5]. Among the polymers with these properties, poly-lactic acid (PLA), poly-glycolic acid (PGA), poly (lactic-co-glycolic acid) (PLGA), and polycaprolactone (PCL) are highly relevant [6].

PLA is an organic acid of natural origin. It is derived from renewable products, such as sugar cane, potato, and corn starch, through the bacterial fermentation of carbohydrates or by chemical synthesis. It is a polymer that can be used to produce absorbable and biodegradable dressings as nanoparticles on the nanometer scale, because as cells grow and organize, the polymer degrades and is absorbed by the body, leading to a natural replacement of the tissue [7]. Nanotechnology, which encompasses nanoparticles, nanofibers, and small biomaterials, is used for the topical administration of wound healing medications.

In recent decades, the encapsulation of bioactive natural products has been used to promote a reduction in wound healing time. Studies in the literature indicate that propolis is a substance of complex composition. It is formed from a gummy and balsamic material that is collected by bees from buds, tree exudates, waxes, and other parts of plant tissues. It is altered in the hive by the addition of a salivary secretion [8]. Its chemical composition depends on variables such as the flora of the region, seasonality, and locations. In addition, it depends on the technique used for production and the different genetic characteristics of the bees. The resinous fraction is rich in phenolic substances, such as phenolic acids, and there is a predominance of flavonoids [9]. Red propolis is classified as the 13th subtype of Brazilian propolis; it was found in hives located along the coast and mangroves in the states of Sergipe, Alagoas, Paraíba, and Bahia [4]. Its botanical origin is mainly due to the plant *Dalbergia ecastophyllum* (L.) Taub. (Fabaceae), which has a reddish resin rich in isoflavones [10].

Propolis is a natural product with high medicinal potential in both human and veterinary medicine. Over the last 100 years, the scientific literature has revealed its pharmacological properties, such as antibacterial [4], anti-inflammatory [11], antifungal, antioxidant [12], and antitumor activity, with its uses for wound healing and tissue repair [13].

Given these perspectives, the present study proposes to develop a hydrophilic polymeric membrane impregnated with nanoparticles of red propolis from Alagoas encapsulated with PLA, with possibilities of application for wound healing.

## 2. Results

### 2.1. The Yield of Red Propolis Extract

The red propolis hydroalcoholic extract obtained showed organoleptic characteristics corresponding to crude propolis. It has a reddish color and a balsamic aroma. The recovery of soluble solids was approximately 25.7%.

### 2.2. Nanoparticles and the Production of Membranes Loaded with Red Propolis Extract

Table 1 summarizes the components present in the prepared membranes. When compared to the pure PLA polymer, the NPs and NPRP obtained showed an opaque white and reddish color, respectively. Furthermore, a decrease in the apparent density of the products was observed, as shown in Figure 1.

The polymeric membrane MNP (membrane impregnated with PLA nanoparticles) and MNPRP (membrane impregnated with nanoparticles loaded with red propolis) were obtained and presented macroscopic characteristics such as uniformity, rigidity, and color (opaque white for MNP and pink (MNPRP) for the red propolis).

### 2.3. Determination of Total Phenols and Total Flavonoids Content

Table 2 presents the content of total phenols, total flavonoids, antioxidant activity using DPPH quantified in the samples of RPE, NPRP, and MNPRP, and were quantified respectively 240.3, 191.7, and 183.4 mg EAG g^−1^ for phenolic compounds and 37.8, 35.8, and 26.4 mg EQ g^−1^ for total flavonoids, respectively in RPE, NPRP, and MNRP.

According to the literature, the red propolis from Alagoas, classified as subtype 13, has varying levels of phenolic compounds above 90 mg g^−1^. As observed in some studies, the phenolic compounds were 95.8 [14], 416.3 [15], and 481.59 mg EAG g^−1^ [16]. On the other hand, red propolis has been observed to have a lower percentage of flavonoids when compared to total phenolic compounds [17]. A study reported by Nascimento [14] highlights the content of 35.1 mg EAG g^−1^ of flavonoids. In other studies, the flavonoid values were 25.0 [10], 32.91 [15], 140.2 [18], and 186.9 mg EAG g^−1^ [16], showing a wide range of total flavonoid amounts registered in the literature.

### 2.4. Antioxidant Activity Using DPPH and FRAP

At the maximum concentration of the extract (25 µg mL^−1^), RPE showed 92.8% ± 0.1% inhibition of the DPPH radical, and an IC_50_ of 13.1 µg mL^−1^. For NPRP, the percentage inhibition of the DPPH radical was 75.8% ± 3.9 and the IC_50_ was 16.1 µg mL^−1^. For MNPRP it was 67.8% ± 1.0, and the IC_50_ was 18.4 µg mL^−1^. In different studies conducted by Lopez [19] and Mendonça [20], they observed that red propolis from Alagoas (G13) showed antioxidant activity with different IC_50_ values at the concentrations of 8.0 and 5.5 µg mL^−1^, respectively.

For FRAP, a hydroalcoholic extract (i.e., RPE), obtained by ultrasound, and NPRP were evaluated, showing high antioxidant capacity. 

### 2.5. Determination of the RPE Chemical Markers

Figure 2 and Table 3 illustrate the chromatographic profile (UPLC-DAD) of red propolis flavonoids identified in the hydroalcoholic extract and NPRP. The identification of chemical compounds was performed by direct comparison with analytical standards, and it was based on the chromatography retention time and the maximum wavelength (λ) in the UV-DAD spectrum. The results obtained reveal the major compounds that are the main markers of red propolis, such as liquiritigenin, daidzein, isoliquiritigenin, formononetin, and Biochanin A, corroborating Gomes et al. [21], who also identified these markers in the ethanol extract of red propolis.

### 2.6. Encapsulation Efficiency (% EE)

Five chemical markers representing the major flavonoids found in the samples were used to determine the content and (%) efficiency of encapsulation. It was observed that Biochanina A showed greater loss (−19.9%) during the process of obtaining the nanoparticles. The encapsulation efficiency obtained in this work was high (80.1–92.8%), highlighting better recovery for the marker daidzein. In general, considering that formononetin is the major compound in red propolis extract, we can infer that the encapsulation efficiency was 85.4% (Table 4).

This is a higher percentage when compared to the work by Pandey [22], who obtained an encapsulation efficiency of 65% when obtaining PLA nanoparticles loaded with quercetin. Likewise, in percentage terms, the encapsulation efficiency could be observed in the findings of the antioxidant evaluation and phenolic compound contents as well as in the total flavonoid contents, which were similar to that of RPE.

### 2.7. Particle Size, Polydispersion Index, and Zeta Potential

Table 4 lists the membrane characterization parameters such as particle size, polydispersity index, and zeta potential. Different particle sizes are observed in NP (194.1 nm) and NPRP (361.2 nm) particles (Figure 3). The nanoparticle suspensions obtained without propolis and with propolis presented, macroscopically, a homogeneous, opaque appearance with white and reddish colors, respectively. The polydispersion index generates information regarding the homogeneous distribution of the particles. The findings of this study show that the NP and NPRP (Table 4) presented satisfactory indices with values lower than 0.3, presenting a homogeneous distribution of the particles, which is characteristic of a monodisperse system.

A slight improvement in this parameter was observed after the incorporation of red propolis into the polymeric matrix. The zeta potential obtained for the nanoparticles showed a negative charge with values of −11.2 and −16.4 for NPs and NPRP, respectively. The zeta potential of the nanoparticles was negative due to the presence of terminal carboxylic groups in the polymers, demonstrating a high potential value [23].

### 2.8. Differential Exploratory Calorimetry

The DSC heating curves are summarized in Figure 4 (nanoparticles) and Figure 5 (membranes). The curve of pure PLA in the form of pellets showed three thermal transitions: glass transition (Tg), crystallization (Tc), and melting temperature (Tm). Thus, it was possible to observe temperatures of 64.38 °C (Tg), 113.8 °C (Tc), and discrete double melting points of 148.5 °C (Tm_1_) and 159.2 °C (Tm_2_), characterizing the glass transition and exothermic and endothermic events, respectively. The values obtained were within the typical ranges reported for PLA, with Tg values between 50 and 80 °C and Tm values between 130 and 180 °C; this corroborates with Auras [24]. After exposure of the PLA to acetone and the addition of PEG and Tween 20, a reduction in the Tg from 64.38 (pure PLA) to 57.78 °C was observed along with the absence of crystallization peaks and a reduction in the Tm_1_ to 145.2 °C and in the Tm_2_ to 149.9 °C in the NPs. The NPRP showed a greater reduction in the Tg from 64.3 to 56.5 °C and in the Tm_1_ to 144.2 °C and the Tm_2_ to 149.4 °C. Double melting peaks (i.e., Tm_1_ and Tm_2_) were observed in the DSC curve of the NPs and NPRP with different orientations when compared to the pure PLA. The different orientation was linked to a decrease in the thermal stability of less-perfect crystals (α’-shaped crystals), which may have weakened their structural organization after exposure to solvent and plasticizer [25]. Figure 4 shows the profile of the heating curves comparing PLA (pure), RPE, NPs, and NPRP, and it highlights the incorporation and miscibility of red propolis in PLA, when observing the absence of traces of the propolis heating curve, corroborating the FTIR findings.

The heating curves of the DSC of the Base M (base membrane) revealed an endothermic peak of NaCMC due to the fact of its higher percentage in the formulation. When comparing the three membranes, Base M (no nanoparticles and no propolis), MNP, and MNPRP, a first-order event characterized by mass loss from water evaporation was observed. This event was accentuated in membranes containing PLA nanoparticles, suggesting that the glass transition of PLA nanoparticles occurs in parallel with evaporation. However, there was a temperature shift leading to the right glass transition peaks of 78.2 and 81.0 °C for MNP and MNPRP, respectively.

Characteristic peaks of the melting temperature were identified in both NPs and NPRP, and they suggest the existence of a similar profile in MNP and MNPRP. The melting temperature of MNP and MNPRP was 146.0 and 145.4 °C, respectively. These are lower temperatures but similar to those found for the nanoparticles, showing the homogeneous aspect of the membrane impregnated with PLA nanoparticles loaded with red propolis.

### 2.9. Thermogravimetric Analysis

In this work, the TGA technique was used to evaluate the stability of RPE, PLA (pure), NPs, and NPRP. Figure 6 shows the thermogravimetric profile of the samples, and Table 5 lists the beginning and end of the thermal event with data on the initial and final temperature of degradation.

Pure PLA was the material that showed high thermal stability, showing only one thermal event starting at 343.7 °C and ending at 382.7 °C with mass groups of 98.1%, associated with ester loss by depolymerization, as described in the literature [26,27]. The thermogravimetric curve of red propolis shows three thermal events.

Event I had an initial event beginning at 88.8 °C and ended at a final temperature of 118.8 °C. It was characterized by the output of water (5.9%) adsorbed on the hydroalcoholic extract of propolis. The second event started at 278.0 °C and ended at 353.4 °C. It may be related to the degradation of phenolic compounds. Event III was marked by an initial temperature of 427.4 °C and a final temperature of 503.6 °C, indicating the emergence of products generated during the thermodegradation process that need to be subjected to higher temperatures for mass loss to occur. The thermogravimetric curve obtained in the NPs’ analysis revealed a profile with high thermal stability, like that found for pure PLA (Table 5). The nanoparticles loaded with red propolis presented two thermal events. The first event showed a reduction in the stability of the polymer after the incorporation of propolis, and the second revealed a loss of mass in the thermodecomposition of PLA and the major components of red propolis.

The thermogravimetric evaluation of the membranes (i.e., Base M, MNP, and MNPRP) is described in Figure 7 and Table 6. The polymeric membrane Base M presented three thermal events. Event I showed an initial and final temperature of 37.43 and 113 °C, respectively. At 73 °C, there was a rapid loss of mass (15.9%) coming from the water adsorbed in the polymer matrix.

The second event was marked by a loss of mass in a short temperature increment (189.6–206.3 °C), indicating the beginning of membrane decomposition, with an unstable stage allowing for greater degradation between temperatures of 258.9 and 336.8 °C, with a mass loss greater than 50%, indicating oxidation and membrane decomposition. The residual mass (ash) for this sample represented 23.4%. This was attributed to the sodium portion of the sodium carboxymethyl cellulose, which has high thermal stability. The membrane impregnated with the PLA nanoparticles presented four thermal events. In Event, I, the volatilization of water adsorbed in the polymeric matrix of the membrane was noted. Events II, III, and IV were attributed to the loss of volatile components, decomposition of NaCMC monomers, and the thermodecomposition of the PLA polymer associated with the loss of ester groups, respectively. The membrane impregnated with nanoparticles loaded with red propolis showed four thermal events and characteristics that were similar to MNP. However, Event III presented greater mass loss, which may be related to an increase in degradation products from red propolis.

### 2.10. ATR Coupled to Fourier Transform Infrared (ATR-FTIR)

The FTIR spectroscopy technique is a sensitive technique that allows for the evaluation of molecular interactions. In this study, interactions between PLA, red propolis, and base membrane (Base M) were compared. Figure 8 and Figure 9 show the spectra obtained for PLA, RPE, NP, NPRP, Base M, and MNPRP. The PLA regions of interest were the bands at 2950 and 2945 cm^−1^, which represent the asymmetric stretching vibrations of CH and CH_3_, respectively [28]. An intense band was observed at 1747 cm^−1^. This is a characteristic peak that represents the vibration of the carbonyl C=O of the ester bonds [29]. The band observed at 1456 cm^−1^ can be attributed to methyl/methylene deformation. The bands at 1184 and 1078 cm^−1^ represent the stretching vibrations of the C-O of the carboxylic groups and the stretching vibrations of the C-O-C [28].

The bands identified in the RPE showed O-H stretching of the phenolic compounds at 3372 cm^−1^ (phenolic hydroxyl group), and the band at 2932 cm^−1^ was related to the alcohol portion of the phenolic substance. The bands at 1618, 1508, and 1448 cm^−1^ corresponded to the C=C stretching of the aromatic ring, and the 1030 cm^−1^ band was attributed to the C-O aromatic ether linking stretch (for flavonoids). The band at 836 cm^−1^ corresponded to the angular deformation outside the aromatic C-H plane and corroborates the findings in the literature [9].

The nanoparticles (NPs) and NPRP showed bands with small displacement and lower intensity when compared to PLA. A characteristic PLA band was detected at 1755 cm^−1^ (C=O carbonyl stretching vibration of the ester bonds), and the bands at 1359 and 1086 cm^−1^ represent the C-O stretching vibrations of the carboxylic groups and vibrations of C-O-C stretching. The membranes were also evaluated by the FTIR technique. Figure 9 shows the spectrum of bands obtained for the Base M. The main bands identified came from NaCMC functional groups due to the fact of their greater proportion in the formulation.

The intense and broadband at 3302 cm^−1^ refers to O-H axial stretching and the formation of intramolecular/intermolecular hydrogen bonds. In addition, it may be associated with a junction of the absorption of this group that is present both in NaCMC as well as in citrus pectin, which is due to the existence of the galacturonic acid chain.

At 2937 cm^−1^, C-H axial deformation occurred. The band at 1743 cm^−1^ refers to the axial strain of -COOH. At 1597 and 1417 cm^−1^, important bands involved in the symmetric deformation of the carboxylate anion (COO-Na^+^) and asymmetric deformation of the carboxylate anion (COO-Na^+^), respectively, were identified. The bands at 1225 cm^−1^ corresponded to the C-O axial strain. The band at 1018 cm^−1^ was due to the fact of C-O-C stretching and the C-C stretch. A band at 895 cm^−1^, referring to C-H angular deformation, was also found, corroborating other authors [30,31].

## 3. Discussion

Currently, the literature reports different types of polymeric dressings to coat and accelerate tissue regeneration. Among the main ones, there are foam dressings, films, hydrogel, hydrogel-based membranes developed with alginate, pectin, CMC, hydrocolloids, and other types of polymers such as PLA (Table 7) and have advantages and disadvantages inherent to the type of polymer matrix [32]. Hydrogel-based membranes arouse immense interest, their high swelling capacity favors the absorption and retention of exudate, controlling the amount of fluids under the wound, maintaining adequate wettability and humidity, supporting the proliferation of fibroblasts, and the migration of keratinocytes [33].

In the last decade, polymeric membranes have attracted much attention due to the fact of their great potential for application in several areas such as biomedical and pharmaceutical applications and in the field of biotechnology. The literature reveals that membranes loaded with bioactive components have shown wound healing potential. In addition, the association with polymeric, synthetic, biodegradable, and bioabsorbable nanomaterials, such as PLA, has shown promise for tissue recovery [5]. In this study, a topical membrane formulation was developed based on NaCMC impregnated with PLA nanoparticles loaded with Brazilian red propolis, and it was characterized in terms of its physicochemical and thermal aspects.

The membrane system developed here presents a differential concerning the different types described in the literature. We performed the incorporation of nanoparticles loaded with red propolis from Alagoas to improve the mechanical and thermal stabilities as a function of the swelling and vapor exchange capacity, as described in the literature [32]. In addition, the membrane became functional by presenting the release of the nanoparticles and conferring a significant antioxidant activity based on the RPE. It is suggested that the biological properties of red propolis are also present in the nanoparticles with RPE. There is no development in the literature on this type of membrane system (CMC + PLA nanoparticles + RPE). In the second stage of this project, mechanical and biological tests will be able to confirm these properties. Significantly, PLA nanoparticles, even in the absence of RPE, could accelerate the healing process. In a comparative study carried out by Bi et al. [39] using PLA and PLA/PVA/SA electrospun fiber membranes for wound healing in vitro and in vivo, the authors observed that isolated PLA fibers allowed for the acceleration of wound healing in rats. The developed membrane system will be able to act synergistically between the hydrophilic membrane (promoting exudate absorption) [32], PLA (protein deposition) [39], and RPE (antioxidant, anti-inflammatory, and healing activity) [4]. The analyses described in this work reveal that the developed membrane has characteristics suitable for its use in in vitro and in vivo biological assays.

The UPLC-DAD analysis of the Brazilian red propolis identified different classes of flavonoids such as isoflavone (daidzein, formononetin, and Biochanina A) chalcones (isoliquiritigenin), and flavone (liquiritigenin), and this corroborated the literature [4]. In this study, formononetin was the major flavonoid. The literature reports antinociceptive and anti-inflammatory effects in rodents using Brazilian red propolis extract and formononetin [41]. Biochanin-A showed antiproliferative and anti-inflammatory activity [42], and isoliquiritigenin and liquiritigenin have a potential antimicrobial effect [41]. In a previous study by this group, Brazilian RPE and membranes showed antimicrobial activity against strains of *Staphylococcus aureus* and *Staphylococcus epidermidis.* Interestingly, RPE showed inhibition of antigen-induced mast cell degranulation, indicating that it has no allergenic potential [4]. Thus, the red propolis extract from Alagoas presents favorable characteristics for incorporation into polymeric membranes for therapeutic applications.

Variations in the levels of phenolic compounds and total flavonoids in Brazilian red propolis obtained by UPLC-DAD and colorimetric assays are directly related to its botanical origin, collection site, seasonality, and extractive method [8]. Due to the great variety of these compounds in Brazilian propolis, the Ministry of Agriculture, Livestock, and Supply regulates that the beekeeping product in the form of the extract must present, using identity and quality fixation, at least 0.25% (*m*/*m*) flavonoids and 0.50% (*m*/*m*) phenolics [43]. The samples evaluated in this work were consistent with the standards established by the legislation regarding these two parameters. Furthermore, there is a similarity between the levels of total phenols quantified in the RPE and the NPRP as well as the levels of total flavonoids. We can infer that the process of obtaining the nanoparticles did not change the phytochemical profile of the RPE.

Due to the fact of its ease and speed, the 2,2-diphenyl-1-picrylhydrazyl (DPPH) reagent has been widely used as a method to evaluate the antioxidant activity of several compounds [8]. The result obtained in this work confirms the literature on the high antioxidant capacity of Brazilian red propolis extract (RPE) [9,20]. The evaluated concentration of NPRP and MNPRP showed a loss of antioxidant activity of 18.3% and 26.9%, respectively, when compared to RPE. This suggests that these losses were related to the low compatibility between the reaction solvent ethanol (polar) and the nanoparticles of the PLA (nonpolar) loaded with red propolis. Similarly, the membrane polymer matrix (NaCMC) was not solubilized by ethyl alcohol, making it difficult to extract the red propolis contents present in the nanoparticles, interfering with the results of the analysis. However, the free radical scavenging power of NPRP incorporated into the membrane matrix was satisfactory, allowing for its application as an antioxidant product.

Assessing the antioxidant activity of propolis by more than one methodology is important, since discrepant values are found in the literature. These discrepancies may be associated with methodological changes and mainly with issues of seasonality, place of material collection, and the types of flora that surround the swarms of bees. Fe^3+^ reduction is often used as an indicator of electron-donating activity, which is an important mechanism of phenolic antioxidant action. In this work, RPE showed relevant antioxidant activity and corroborates the findings in the literature with 1472.8 µmol Fe^2+^ of dry extract [44], indicating that the red propolis extract was a very strong antioxidant material. Oldoni et al. [45] found 259.30 ± 9.50 µmol Fe^2+^ g^−1^ of dry weight for optimized propolis samples from the state of Paraná, Brazil. Andrade et al. [46] reported a FRAP value of 633.1 µmol TE g^−1^ of dry weight for the Brazilian propolis extract.

Nanoparticles are being used on a large scale as a drug delivery system. They play an important role in this regard, and their size significantly influences their release profile. Thus, among the characterization parameters, particle size is significant in the aspects of the degradation and release, biodistribution, and absorption of nanoparticles [47,48]. Studies in the literature state that the particle size generally depends on the target tissue; however, maximum sizes are in the range of 20–400 nm [48]. In this work, it was possible to obtain satisfactory sizes between 194.1 (NPs) and 361.2 nm (NPRP). This suggested that the larger size of the NPRP came from the entrapment of red propolis within the polymeric matrix.

The DSC heating curves allowed us to observe the emergence of double melting peaks that were identified in the pure PLA, a common phenomenon described in the literature. It is associated with a step in the production of the polymer and is related to changes in crystal growth. This is due to the existence of polymorphism, with the simultaneous presence of two populations of crystals that developed with different sizes and stability, making it possible to reach the melting state at a lower temperature followed by a structural reorganization of the crystal, resulting in a perfect or a larger crystal stability during the second fusion event [49,50,51]. The decrease in the Tg observed in the NPs and NPRP was directly related to an increase in the mobility of PLA chains, facilitating its folding into a crystal lattice with the addition of plasticizing agents and other compounds that interfere with the plasticity of the polymer. Thus, it is suggested that the addition of PEG and Tween 20 to form the nanoparticles was responsible for reducing the glass transition of the PLA. In a study on amorphous poly(lactic acid) crystallization induced by acetone vapor to form a specimen of high crystallinity and transparency, it was observed that the solvent was essential for the reduction in the Tg and Tms, in addition to its transparency and opacity [52]. Thus, it was observed in the present study that the reduction in Tg may also be associated with exposure to acetone.

The thermogravimetric curve obtained in the NPs’ analysis revealed a profile similar to that found for pure PLA (Table 5), with high thermal stability. However, a lower stability was observed when compared to pure PLA, since the degradation occurred at a lower temperature. In addition, this behavior corroborates the findings in the exploratory scanning calorimetry, which revealed a smaller glass transition, which is a parameter that confers greater stability and resistance to the polymer. The nanoparticle loaded with red propolis presented two thermal events, showing the reduced stability of the polymer when compared to pure PLA. On the other hand, the high thermal stability of PLA provides greater stability to propolis, favoring the preservation of its constituents. The second event stands out, showing a loss of mass in the thermodecomposition of PLA and the major components of red propolis.

The membrane impregnated with the PLA nanoparticles showed four thermal events. In the first event, it is possible to observe a mass loss associated with the volatilization of water adsorbed in the polymeric matrix of the membrane, with a lower mass loss (9.7%) when compared to the Base M. Events II, III, and IV were attributed to nonstable multistage mass loss involving the decomposition of volatile components and the decomposition of NaCMC monomers. Event IV was traced by a thermodecomposition of the PLA polymer associated with the loss of ester groups by decompacting depolymerization, which is widely described in the literature [26,27,53]. MNP also showed residual mass from sodium mineral salt however in a lower proportion (20%) when compared to Base M. The thermal profile and residual percentage of MNPRP were similar to MNP.

The FTIR spectroscopy technique allowed for the identification of the amorphous and crystalline phases of the pure PLA, with the absorption bands at 871 and 754 cm^−1^, respectively, and this corroborates the literature [54]. The nanoparticles loaded with red propolis (NPNALRP) showed a high similarity to the NPs. In this way, the bands from red propolis were suppressed, contributing to the DSC data and confirming the encapsulation of propolis by the polymer. MNPRP FTIR analysis revealed a similarity to the MNP, with small shifts and lower transmittance intensity. However, the band at 1743 cm^−1^ showed higher intensity. Possibly, this band was more expressive due to the stretching vibration of the carbonyl C=O of the ester bonds of the nanoparticles and NaCMC-Na, allowing us to infer the homogeneity of the MNPRP obtained.

## 4. Materials and Methods

### 4.1. Materials

Sodium carboxymethylcellulose (NaCMC) 3000 (Denver Especialidades Químicas, São Paulo, Brazil), citrus pectin (Êxodo Científica, São Paulo, Brazil), and propylene glycol (Synth, São Paulo, Brazil) were used in the production of the membranes. The polylactic acid (PLA) Mw 75,000 was acquired from Biomater (São Paulo, Brazil); the acetone, polyoxyethylene 20 sorbitan monolaurate, and polyethylene glycol 6000 were acquired from Synth (São Paulo, Brazil). The Folin–Ciocalteu reagent (Êxodo Científica, São Paulo, Brazil), 2,2-diphenyl-1-picrylhydrazyl radical (DPPH), gallic acid, quercetin, and 2,4,6-tris(2-76 pyridyl)-s-triazine were purchased from Sigma Aldrich (Steinheim, Germany). The flavonoids, namely, daidzein, formononetin, and biochanin A, were acquired from Sigma-Aldrich (St. Louis, MO, USA). Liquiritigenin and isoliquiritigenin were acquired from Extrasynthese^®^ (Lyon Nord, France) and were used as the analytical standards. High-performance liquid chromatography (HPLC) grade methanol was purchased from J.T. Baker (Mallinckrodt, Mexico), acetonitrile was purchased from Fisher Scientific (Leicestershire, UK), and the Milli-Q grade water was produced in a lab. Sodium carbonate was supplied by Vetec Quíımica Fina (Rio de Janeiro, Brazil). Anhydrous monobasic sodium phosphate (Neon Commercial, São Paulo, Brazil) was used in the phenol release test.

### 4.2. Red Propolis Extract (RPE)

The red propolis (RP) was obtained from the apiary located in the mangrove region of the municipality of Marechal Deodoro-Alagoas, Brazil (S9°42′10.2924″ and W35°54′21.5316″). The crude propolis (50.0 g) was transferred to a 250 mL volumetric flask, and a volume of 166.7 mL of 80% ethyl alcohol was added. Then, the mixture was taken to the sonicator (UltraCleaner 750-Unique) with a constant frequency of 25 kHz and power of 100 W for 30 min. The extract was filtered through filter paper and placed in an amber glass bottle until analysis.

### 4.3. Preparation of the PLA Nanoparticles Loaded with Red Propolis (NPRP) and Membrane

#### 4.3.1. Preparation of the PLA Nanoparticles Loaded with Red Propolis (NPRP)

Obtaining the nanoparticles occurred by emulsification and diffusion of the solvent. Approximately 300 mg of PLA polymer was dissolved in 15 mL of acetone in a closed bottle and taken to a water bath at 70 °C for 40 min. At room temperature, the PLA solution was vigorously stirred on a mechanical stirrer at 1000 rpm, followed by the addition of polyoxyethylene 20 sorbitan monolaurate (136 µL), stirred for 60 s. After this step, 21 mg of RPE was incorporated into the PLA solution until the solution was homogenized. Then, 10 mL of an aqueous solution of Polyethylene Glycol 6000 (PEG 1% *w*/*v*) was added dropwise to form a suspension of nanoparticles by precipitation and diffusion of the solvent. The suspension remained under continuous agitation in an oven with a temperature of 30 °C and circulation of renewed air for 18 h hours for residual removal of the organic solvent. The obtained nanoparticle was frozen for 10 min in liquid nitrogen and dried by lyophilization for 24 h (Terroni LS 3000 freeze dryer, São Carlos, São Paulo, Brazil). The dried nanoparticles were stored in a desiccator protected from light until further studies. The same protocol was also followed for the synthesis of the nanoparticle without propolis (NPs).

#### 4.3.2. Development of the Polymeric Membrane Loaded with NPRP

For the preparation of the polymeric membrane, the components of the formulation, sodium carboxymethyl cellulose, citrus pectin, and propylene glycol 400, were weighed on an analytical balance (Table 1). In a beaker previously containing 70 mL of distilled water, the formulation components were added. The mixture was stirred in a mechanical shaker at 1700 RPM until homogenization and then distilled water was added up to 100 mL. Then, 100 mg of PLA nanoparticles loaded with red propolis and without propolis were incorporated into 10 mL of the base. The homogenized content was placed in glass plates (inhouse manufacture) and placed in an oven at 37 °C for 48 h.

### 4.4. Size, Polydispersity Index, and Zeta Potential of the Nanoparticles

The average particle diameter and the polydispersion index were determined using the dynamic light scattering technique. Individually, a suspension of the nanoparticle with and without propolis was prepared at a concentration of 400 µg/mL in Milli-Q water. From this suspension, a 5-fold dilution was made in Milli-Q water and analyzed in triplicates of 64 scans of Zetasizer apparatus model Nano-ZS from Malvern (Worcestershire, UK). The zeta potential was obtained using the electrophoretic mobility method in a Zetasizer apparatus and the triplicate result was expressed in millivolts (mV).

### 4.5. Rupture of the Nanoparticles Loaded with Red Propolis (NPRP)

To quantify the contents of flavonoids (TFC) and total phenols (TPC) as well as to evaluate the antioxidant activity by the method of DPPH, it was necessary to dissolve the nanoparticles in chloroform, obtaining a clear solution of 1 mg mL^−1^. After that, the solution was stored in an amber flask until analysis.

### 4.6. Extraction of the Red Propolis Impregnated as Nanoparticles in the Polymeric Membrane

The polymeric membrane impregnated with the nanoparticles loaded with red propolis was weighed whole on an analytical balance (corresponding to 7 mg of RP) and was ground in a mortar and solubilized in 10 mL of 50% alcohol in an ultrasonic bath for 1 h. Obtaining a theoretical concentration of 0.7 mg mL^−1^ was used to evaluate phenols, flavonoids, and antioxidant activity.

### 4.7. Total Phenol Content (TPC)

The total phenol content was quantified according to the colorimetric method using Folin–Ciocalteu Reagent (FCR) described by Xavier [55], with modifications. From the stock solution of RPE (4 mg mL^−1^), suspension of 1 mg mL^−1^ of NPRP, and 0.7 mg mL^−1^ of MNPRP, quantification was performed as follows: It was diluted into vials of 5 mL with concentrations of 20 µg mL^−1^. An aliquot of the samples was added to a 5 mL volumetric flask, containing 3 mL of deionized water, 0.4 mL of RFC, and 0.6 mL of saturated 20% sodium carbonate solution. It was filled with water until the final volume and stirred for another 15 s. The reaction took place in the dark for 20 min in a thermostatic bath at 40 °C. The absorbance was measured using a Shimadzu U-VIS spectrophotometer at a wavelength of 760 nm against a blank consisting of FCR solution, sodium carbonate, and deionized water. A calibration curve was constructed from the analysis of different concentrations of gallic acid, varying from 2.0 to 9.0 µg mL^−1^, and its data on absorbances based on calibration equation: y = 0.1226x + (−0.0176) and r^2^ = 0.9971. The results are expressed in terms of milligrams of gallic acid equivalent (GAE) per g propolis.

### 4.8. Determination of the Total Flavonoid Content (TFC)

The total flavonoid content was quantified according to the colorimetric method using aluminum chloride described by XAVIER [55], with modifications. From the stock solution of RPE (4 mg mL^−1^), suspension of 1 mg/mL of NPRP, and 0.7 mg mL^−1^ of MNPRP, quantification was performed as follows: It was diluted to 5 mL flasks with concentrations of 200 µg mL^−1^. An aliquot of the sample was added to a 5 mL volumetric flask, containing 3 mL of methanol and 0.1 mL of aluminum chloride solution (5%). It was completed with methanol to the final volume and stirred for 15 s. The reaction took place in the dark for 30 min. The absorbance was measured in a Shimadzu Uv-VIS spectrophotometer at a wavelength of 425 nm against a blank containing only methanol and aluminum chloride. A calibration curve was constructed from the analysis of the different concentrations of quercetin, varying from 2.0 to 14 µg mL^−1^, and its data on absorbances based on calibration equation: y = 0.0615x + (0.011) and R^2^ = 0.9975. The results are expressed in terms of milligrams of quercetin equivalent (QE) per g propolis.

### 4.9. Antioxidant Activity Assay

#### 4.9.1. Evaluation of the Antioxidant Activity by the DPPH Method

The radical scavenging antioxidant activity (RSA) of the RPE, NPRP, and MNPRP was determined using the DPPH assay according to the methods described in the literature [55,56] with a few modifications. A 0.1 mmol L^−1^ solution of DPPH radical (Sigma Aldrich) in ethanol was prepared and stored in amber glass. From this solution, 2 mL was added to 5 mL amber volumetric flasks, and then RPE, NPRP, or MNPRP aliquots in concentrations of 1.0, 5.0, 10.0, and 25.0 µg mL^−1^ were added, respectively. The volume was completed and left to wait for the reaction in the dark for 30 min. After the reaction, readings of the samples were made in triplicate in a UV-Vis spectrophotometer, model Shimadzu 1240, at a wavelength of 517 nm to measure the sequestering capacity of the DPPH radical. A blank containing ethanol and DPPH were used to calculate the percentage of DPPH radical remaining, followed by the formula (% of DPPH remaining = ((AS − AE)/(AD − AE)) × 100), where AS = reaction absorbance the DPPH radical and the sample; AE = absorbance of ethanol; AD = absorbance of DPPH radical diluted in ethanol to a volume of 5 mL. After determining the remaining DPPH radical, the percentage of inhibition of the DPPH radical was determined using the following formula: (% inhibition of the DPPH radical = 100 − % DPPH Remaining).

#### 4.9.2. FRAP Assay

The ferric reducing ability (FRAP) was performed according to the method described in the literature [55]. In 5 mL flasks, 90 µL aliquots containing 25 µg mL^−1^ of the RPE, NPRP, and MNPRP samples; 270 µL of distilled water; 2.7 mL of FRAP reagent (prepared from 25 mL of 0.3 M acetate buffer, 2.5 mL of 10 mM TPTZ (2,4,6-tris(2-pyridyl)-s-triazine) solution, and 2.5 mL of 20 mm). The solutions were homogenized in a tube shaker and kept in a water bath at 37 °C for 30 min. After the reaction time, readings were performed in a UV-Vis spectrophotometer, model 1240 Shimadzu, at a wavelength of 595 nm. The analyses were performed in triplicate, and the results are expressed in µM of ferrous sulfate/g of red propolis.

### 4.10. Identification of RPE Markers Using the UPLC-DAD Method

The identification of the flavonoids contained in the RPE and NPRP was performed using the high-performance liquid chromatography technique coupled with a diode array detector (UPLC-DAD) (model: Shimadzu), described by Mendonça [20]. From a stock solution of 4 mg/mL of RPE, a dilution in chloroform was prepared, using 5 mL volumetric flasks to obtain concentrations of 400 µg mL^−1^, followed by filtration through a 0.22 µm unit filter, and (2 µL) then the sample was injected into the UPLC-DAD to identify flavonoids. For the identification of NPRP flavonoids, a solution in chloroform at an initial concentration of 4 mg mL^−1^ was prepared. Then, to obtain complete solubilization, the sample was left in an ultrasonic bath for 5 min. Next, the solution was filtered through a paper filter, and from there, a dilution was made, taking 1 mL aliquot of the stock solution into a volumetric flask of 10 mL to obtain a concentration of 400 µg mL^−1^. This solution was filtered through a 0.22 µm unit filter, and (2 µL) then injected into the UPLC-DAD.

The UPLC used consisted of the following modules: a high-pressure pump (model: LC-20ADXR), degasser (model: DGU-20A3R), auto-injector (model: SIL-20AXR), chromatographic column oven, diode array detectors (model: EPDM-20A) and fluorescence detector (model: RF-20A), a controller (model: CBM-20A), and Shimadzu Labsolution software. The flavonoids were separated using a reverse-phase column (C 18, 150 × 4.6 mm; 5 µm), a mobile phase consisting of solvent A (Milli-Q water) and solvent B (acetonitrile), pumped at a flow rate of 0.3 mL min^−1^. The initial elution gradient consisted of water (70%) and acetonitrile (30%) with a variation of the percentage of B to 100% in 40 min, maintaining an isocratic condition of solvent B (100%) up to 53 min and returning to condition at 54 min, followed by conditions of isocratic B (30%) up to 60 min. Analytical standards of the flavonoids described as markers (i.e., daidzein, liquiritigenin, pinobanskin, isoliquiritigenin, formononetin, pinocembrin, and biochanin A) were used to identify the flavonoids of RPE and NPRP by comparing the ultraviolet spectra obtained by an array detector photodiodes at different wavelengths (λ: 249, 281, 286, 275, and 366 nm) and the retention times obtained for standards and samples.

### 4.11. Encapsulation Efficiency (%)

The encapsulation efficiency (EE) of NPRP was assessed using UPLC-DAD, considering the quantitative content of formononetin identified in the extract. Initially, 57.143 mg of a solid mass of NPRP was weighed, which theoretically contains 7% of the incorporated red propolis. Then, in a volumetric flask with a capacity of 10 mL, the NPRPs were solubilized in chloroform, obtaining a working solution in the final concentration of 400 µg mL^−1^. To quantify the markers, the same procedure was performed for identification and quantification in the RPE. The encapsulation efficiency (%) was calculated using Equation (1):EE (%) = [QFTN/QFTE] × 100(1)
where EE = encapsulation Efficiency; QFTN = quantity of formononetin determined in the NPRP; QFTE = quantity of formononetin determined in the RPE.

### 4.12. Thermal Analyses (DSC and TG)

The thermal properties of nanoparticles and membrane were determined using differential scanning calorimetry (DSC) and thermogravimetric analysis (TGA) [4]. The DSC curves were performed on a Shimadzu (model: DSC-60), using an aluminum crucible under a nitrogen atmosphere with a flow rate of 50 mL min^−1^ in the temperature range of 25–200 °C, at a heating rate of 10 °C min^−1^, using 2 mg ± 10% of hermetically sealed samples. The equipment was previously calibrated with an Indian standard supplied by Shimadzu.

The thermogravimetric curves of the samples were obtained on Shimadzu equipment (model: TGA-50), using a platinum crucible, under a nitrogen atmosphere with a flow rate of 50 mL min^−1^, between the temperature bands of 25 and 600 °C, with heating rates of 10 °C min^−1^. The initial mass of the analyzed samples was approximately 5 mg ± 10%. The standard used for calibrating the equipment was calcium oxalate recommended by Shimadzu.

### 4.13. ATR- FTIR Spectroscopy

The absorption spectra in the infrared region for the RPE, NPRP, and MNPRP samples in the 4000 to 700 cm^−1^ range were obtained in Nicolet iS 10 spectrometers from Thermo Fisher Scientific in the attenuated total reflectance (ATR) mode, with 4 resolution cm^−1^ and 64 sweeps. The spectra were obtained at room temperature (≈20 °C) with the direct addition of nanoparticles with and without propolis and membrane (in film form) in the device, without previous treatment.

## 5. Conclusions

Nanoparticles loaded with red propolis are presented as an important biodegradable and bioabsorbable product that is rich in phenolic compounds. Their antioxidant activity was proven by colorimetric methods such as DPPH and FRAP. In addition, nanoparticles with adequate sizes were obtained so that they could be used as a substance release system. The characterization of the nanoparticles allowed us to infer that there was encapsulation of the red propolis extract, reflecting satisfactory aspects regarding the quality and quantity of the phenolic and total flavonoid contents. In addition, the thermal and chemical profile of the nanoparticles demonstrated a similarity to the pure PLA polymer, suggesting the thermal and chemical stability of the polymer. Hydrophilic polymeric membranes impregnated with PLA nanoparticles were successfully obtained and showed antioxidant activity. The chemical, thermal, and morphological characterization allowed us to affirm that there was an increased membrane thermal stability. In this scenario, the characteristics of the developed membrane present its strong potential as a new biomaterial for application in the healing of skin wounds. However, complementary tests, such as biocompatibility, biodegradation, and toxicity, must be performed to confirm the development of a new biomaterial.

## Figures and Tables

**Figure 1 molecules-27-06959-f001:**
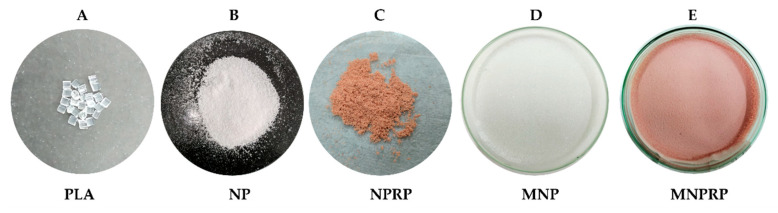
PLA and membranes: (**A**) pure PLA; (**B**) nanoparticles without propolis (NPs); (**C**) nanoparticles with propolis (NPRP); (**D**) membrane impregnated with nanoparticles without propolis (MNP); (**E**) membrane impregnated with nanoparticles loaded with propolis (MNPRP).

**Figure 2 molecules-27-06959-f002:**
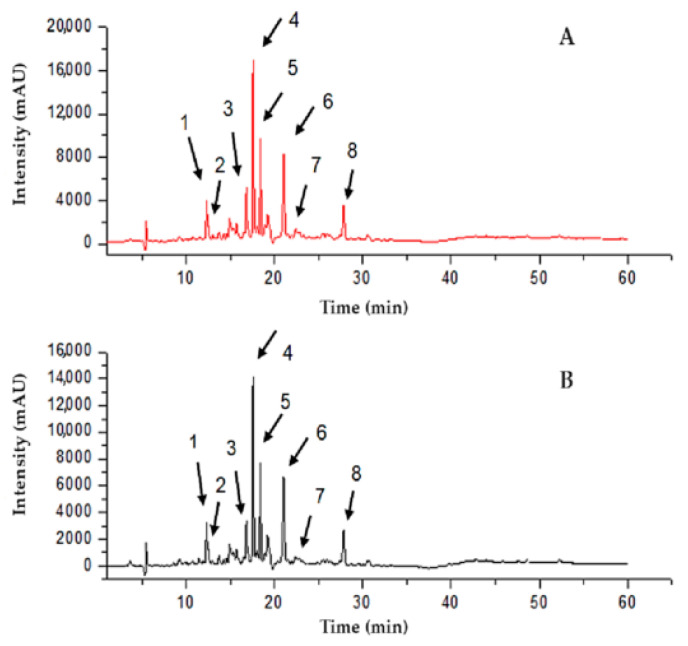
Determination of the markers of red propolis extract (RPE) (**A**) and NPRP (**B**) using UPLC-DAD. Chromatogram of the RPE. 1—Liquiritigenin, 2—Daidzein, 3—Isoliquiritigenin, 4—Formononetin, 5—Unk 1, 6—Unk 2, 7—Biochanin A, 8—Unk 3.

**Figure 3 molecules-27-06959-f003:**
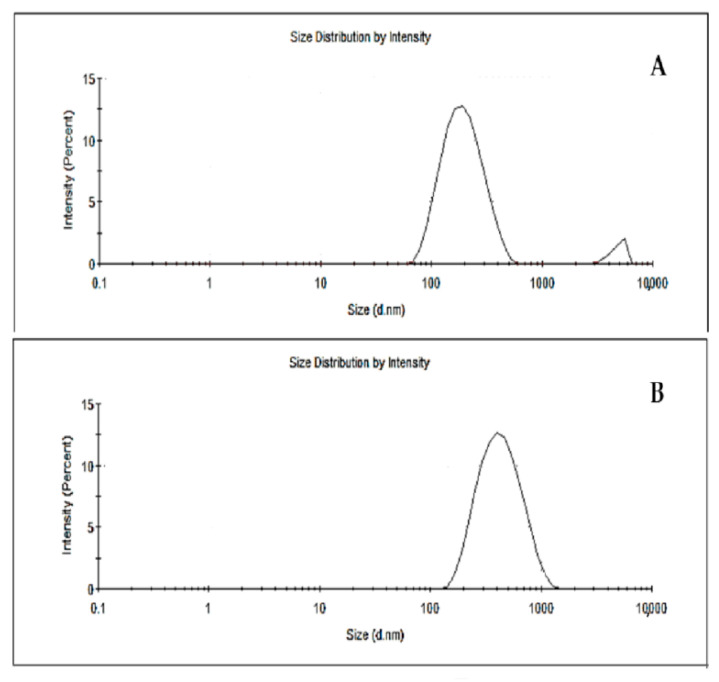
Particle size distribution; (**A**) NPs (nanoparticles without propolis), (**B**) NPRP (nanoparticles loaded with red propolis).

**Figure 4 molecules-27-06959-f004:**
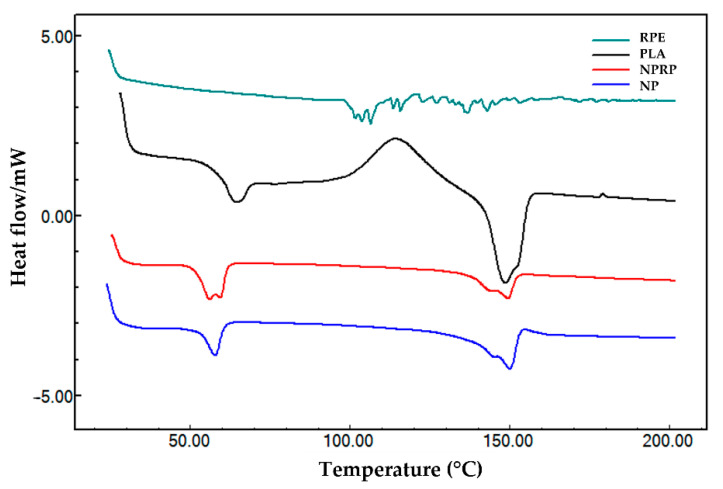
DSC curves of red propolis extract (RPE), PLA, nanoparticles loaded with red propolis (NPRP), and nanoparticles without red propolis (NPs).

**Figure 5 molecules-27-06959-f005:**
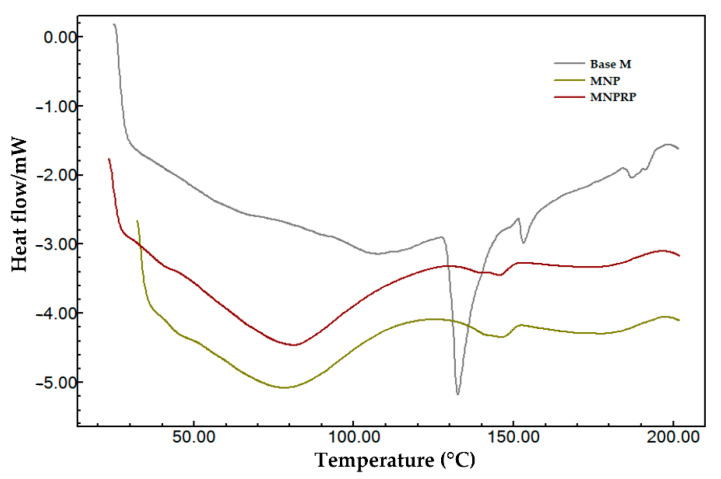
DSC curves of Base M (base membrane-no nanoparticles and no propolis), membrane with NPs, and membrane with NPRP.

**Figure 6 molecules-27-06959-f006:**
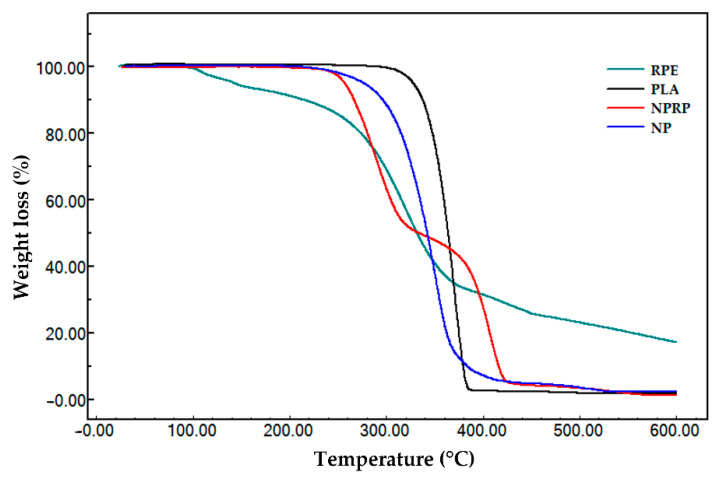
Thermogravimetric profile of red propolis extract, PLA, NPs, and NPRP.

**Figure 7 molecules-27-06959-f007:**
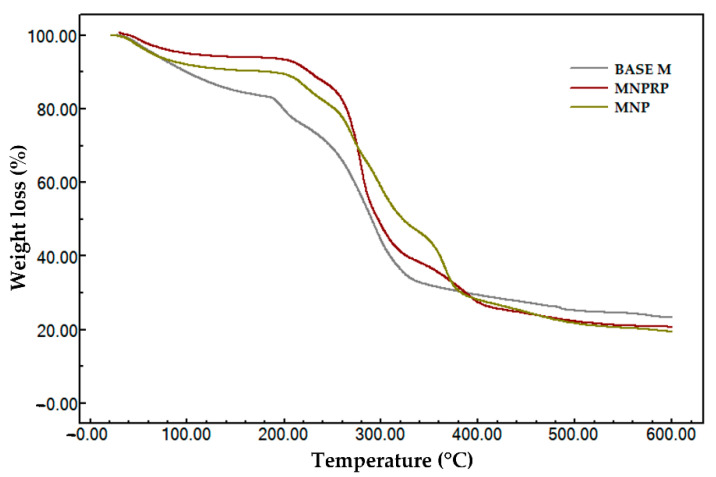
Thermogravimetric profile of membrane: Base M (membrane without propolis and nanoparticles), MNP, and MNPRP.

**Figure 8 molecules-27-06959-f008:**
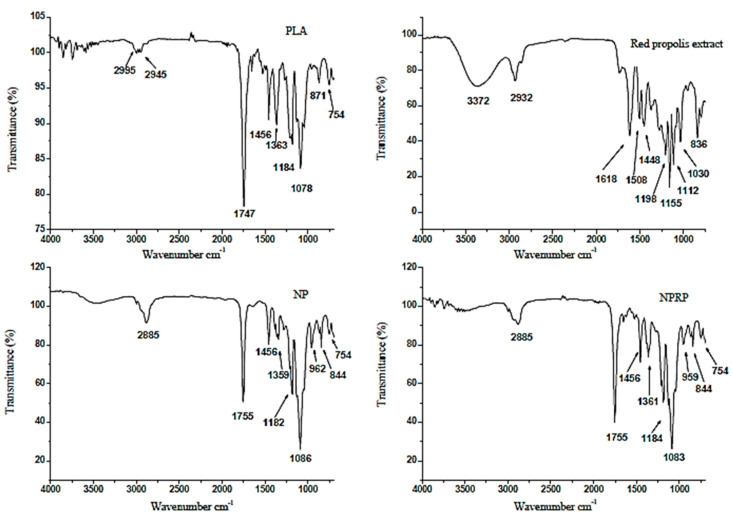
Attenuated total reflectance-Fourier transform infrared spectroscopy (ATR-FTIR) related to the PLA, RPE, NPs, and NPRP.

**Figure 9 molecules-27-06959-f009:**
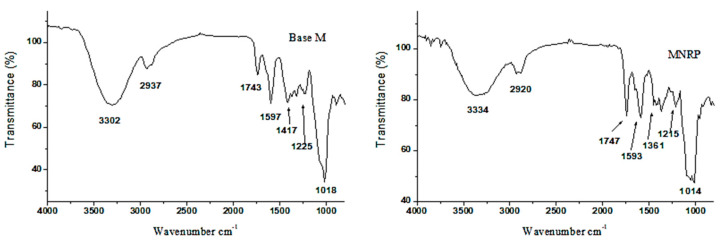
Attenuated total reflectance-Fourier transform infrared spectroscopy (ATR-FTIR) related to the Base M and MNPRP.

**Table 1 molecules-27-06959-t001:** Compositions of the membranes impregnated with nanoparticles.

Composition % (Proportion, *w*/*w*)			
Component	Base M	MNP	MNPRP
NaCMC	0.9	0.9	0.9
Propylene glycol	1.6	1.6	1.6
Citrus pectin	0.6	0.6	0.6
Distilled water	96.9	96.8	96.8
NPs	-	0.1	-
NPRP	-	-	0.1

NaCMC = sodium carboxymethylcellulose; NPs = nanoparticles without propolis; M = membrane; NPRP = nanoparticles with red propolis; MNPRP = membrane incorporated with nanoparticles of red propolis.

**Table 2 molecules-27-06959-t002:** The total content of phenols (TPC) and flavonoids (TFC) of RPE, NPRP, and MNPRP by the Folin–Ciocalteu method, aluminum chloride method, and antioxidant activity by DPPH and FRAP assays.

			DPPH^●^		
Sample	TPC (mg GAE g^−1^) *	TFC (mg EQ g^−1^) **	RSA (%)	IC_50_ (µg mL^−1^)	FRAP (μmol Fe^2+^ g^−1^) ***
RPE	240.3 ± 3.4	37.8 ± 0.9	92.8 ± 0.1	13.1	1400.2 ± 2.6
NPRP	191.7 ± 0.3	35.8 ± 3.9	75.8 ± 3.9	16.1	1294.2 ± 0.5
MNPRP	183.4 ± 2.1	26.4 ± 1.9	67.8 ± 1.0	18.4	696 ± 0.8

* Gallic acid equivalents; ** quercetin equivalent; *** ferrous sulfate equivalent antioxidant capacity. Ferric reducing antioxidant power (FRAP); RSA = radical scavenging antioxidant activity. Values are the mean ± SD.

**Table 3 molecules-27-06959-t003:** Identification of red propolis markers by UPLC-DAD.

			RPE (400 µg mL^−1^)		NPRP (400 µg mL^−1^)		Recovery
Compounds	λ	Peak	RT (min)	Area	RT (min)	Area	(%) *
Liquiritigenin	275	1	12.3	34,750	12.31	28,807	82.9
Daidzein	249	2	12.51	18,096	12.5	16,785	92.8
Isoliquiritigenin	366	3	16.82	71,587	16.81	58,239	81.4
Formononetin	249	4	17.64	120,059	17.63	102,498	85.4
Unk 1	281	5	18.42	79,979	18.41	63,478	79.4
Unk 2	286	6	21.07	109,356	21.06	92,883	84.9
Biochanin A	249	7	23.25	6540	23.21	5239	80.1
Unk 3	281	8	27.83	44,042	27.86	36,366	82.6

λ = wavelength; RT = time retention; Unk = substance present but not identified. * Percentage recovered in the NPRP compared to the content quantified in the RPE at a concentration of 400 µg mL^−1^.

**Table 4 molecules-27-06959-t004:** Particle size, polydispersion index, zeta potential, and encapsulation efficiency of NPs and NPRP.

Nanoparticles	Particle Size (nm)	Polydispersion Index (PDI)	Zeta Potential (mV)	EE ** (%)
NPs	194.1 ± 0.082 *	0.289 ± 0.024 *	−11.2	-
NPRP	361.2 ± 2.71 *	0.239 ± 0.019 *	−16.4	85.4

* Values are the mean ± SD. ** EE = encapsulation efficiency using formononeti; NPs = nanoparticles without propolis; NPRP = nanoparticles loaded with red propolis.

**Table 5 molecules-27-06959-t005:** Thermogravimetric events of RP, PLA, NP, and NPRP.

Sample	Stage(s)	Temperature		Δm (%)
		Onset (°C) *	Endset (°C) **	
RPE	I	88.8	118.8	5.98
	II	278.0	353.4	62.79
	III	427.4	503.6	14.20
	I–III	24.0	600.0	82.96
PLA	I	343.7	382.7	98.19
NP	I	316.3	374.0	97.48
NPRP	I	260.4	312.0	51.55
	II	387.8	444.2	47.18
	I–II	24.0	600.0	98.73

* Onset = onset of degradation; ** endset = end of degradation; % Δm = percentage of mass loss.

**Table 6 molecules-27-06959-t006:** Thermogravimetric events of Base M, MNP, and MNPRP.

Sample	Stage(s)	Temperature		Δm (%)
		Onset (°C) *	Endset (°C) **	
Base M	I	37.4	113.7	15.94
	II	189.6	206.3	7.85
	III	258.9	336.8	52.91
	I–III	24.0	600.0	76.60
MNP	I	33.7	86.6	9.74
	II	206.4	228.9	7.17
	III	260.9	271.8	26.38
	IV	390.0	414.4	37.18
	I–IV	24.0	600.0	80.48
MNPRP	I	40.1	82.5	6.86
	II	211.7	235.7	6.80
	III	265.9	289.6	43.76
	IV	310.1	386.4	22.69
	I–IV	24.0	600.0	80,11

* Onset = onset of degradation; ** endset = end of degradation; % Δm = percentage of mass loss.

**Table 7 molecules-27-06959-t007:** Polymeric wound dressings: advantages and disadvantages.

Type of Polymeric Dressings	Polymer Matrix	Characteristics	Advantages	Disadvantages	Reference
Foam	Polyurethane	Flexible polyurethane foam and lignine nanoparticles enriched with green propolis	Antibacterial activity due to propolis allows for cell adhesion, is hydrophilic, and accelerates healing	Very adherent, forming an opaque layer that makes it difficult to monitor the wound, not applicable to dry wounds and with low stability	[34]
Film	Pectin	Pectin film enriched with allantoin, plasticized with glycerol	Absorbs exudate, flexible, proper contact angle, and accelerates healing	Opaque layer formation, intermediate swelling ability	[35]
Hydrogel membrane	Polyvinyl alcohol (PVA)	PVA membrane enriched with red propolis	Antibacterial activity due to the propolis, high swelling and traction capacities	Variable mechanical and swelling stability, opaque, semipermeable to gases and to water vapor	[36]
	Sodium carboxymethylcellulose (NaCMC)	Sodium carboxymethylcellulose membrane enriched with red propolis	Antibacterial activity due to propolis, high swelling capacity, antiallergic, low cost		[4]
Alginate membrane	Alginate	Alginate and PVA membrane enriched with curcumin nanoparticles	Antibacterial activity due to the curcumin, high swelling capacity, transparent, allows for gas exchange	Swelling, stability, and variable mechanics	[37]
Poly(lactic acid) (PVA) membrane	Poly(lactic acid) (PLA)	Polylactic acid membrane enriched with babassu oil	Formed by pores, it maintains moisture on top of the wound, is nontoxic, promotes cell migration, and has antimicrobial activity	Low swelling capacity, formation of opaque layer	[38]
	Poly(lactic acid) (PLA)/sodium alginate (SA)/poly(vinyl alcohol) (PVA)	PLA/PVA/SA fiber membranes	Mechanical stability, biocompatible and biodegradable, accelerates wound healing		[39]
	Poly(lactic acid)	PLA membrane enriched with green propolis	Mechanical stability, biocompatible and biodegradable, with antimicrobial activity		[40]

## Data Availability

The data presented in this study will be available upon request.

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
