# Peer review of "Development of a Polymeric Membrane Impregnated with Poly-Lactic Acid (PLA) Nanoparticles Loaded with Red Propolis (RP)"

_molecules, 2022, doi:10.3390/molecules27206959_

Round 1
Reviewer 1 Report
I suggest the authors evaluate any in vitro biological functions of the NPs used in this report
What about the solubility and biodegradability of the materials used for this study?
In what kind of direct utilization can be translated by these reported materials?
I expect the author to address clear biofunctional applications in the abstract and conclusion sections?
Author Response
- I suggest the authors evaluate any in vitro biological functions of the N.P.s used in this report.
Answer: Thank you for the critical comment. The goals of the present work are the development and characterization of this new RED PROPOLIS-based biomembrane. It is an essential part of the project. The investigation of the in vitro biological functions is underway and will be published elsewhere.
- What about the solubility and biodegradability of the materials used for this study?
Answer: Polylactic acid (PLA)—based polymers are ubiquitous in the biomedical field thanks to a combination of attractive properties: biocompatibility, along with the fact that their degradation products do not elicit critical responses and are quickly metabolized by the body.
The biopolymer is soluble in organic solvents such as acetonitrile, acetone, chloroform, methylene chloride, dichloroacetic acid, tetrahydrofuran, and others (LASPRILLA et al., 2012).
The process of biodegradation and absorption of PLA in vivo is described in the literature as a long cascade of events. There is a severe irreversible change, gradually compromising its properties when exposed to different environments for a prolonged time. Other mechanisms are involved in the process of degradation and loss of properties. The literature highlights the occurrence of chemical, microbial hydrolysis, photochemical, thermal, and enzymatic degradation, which mainly occur by main chain or side chain scission (AURAS et al., 2004; MULLER, 2008).
Degradation by hydrolysis is the most common path, occurring in two steps. The first is marked, by the non-enzymatic breaking of the ester groups, leading to a gradual reduction in molecular weight, forming oligopolymers (hydrobiodegradable polymers). Sometimes, this step can be faster with acids or bases. The second step is characterized by reduction to form lactic acid monomers, which in turn are used by microorganisms to produce carbon dioxide, methane, and water (AURAS et al., 2004; MULLER, 2012, BARBANTI et al., 2005; FARAH et al., 2016; CASALINI et al., 2019).
The biodegradation behavior of polyester is a fundamental and relevant characteristic, highlighting medical and industrial applications. PLA degradation has been studied in animal and human bodies for medical applications, such as implants, surgical sutures, and drug delivery materials (FARAH et al., 2016).
References
AURAS, R., HARTE, B., & SELKE, S. (2004). An overview of polylactides as packaging materials. Macromolecular Bioscience, 4(9), 835–864. https://doi.org/10.1002/mabi.200400043.
BARBANTI, S. H., ZAVAGLIA, C. A. C., & DUEK, E. A. R. (2005). Polímeros bioreabsorvíveis na engenharia de tecidos. Polímeros, 15(1), 13–21. https://doi.org/10.1590/S0104-14282005000100006
CASALINI, Tommaso et al. A Perspective on Polylactic Acid-Based Polymers Use for Nanoparticles Synthesis and Applications. Frontiers In Bioengineering And Biotechnology, [s. l], v. 7, n. 1, p. 1-16, 11 out. 2019. Frontiers Media SA. http://dx.doi.org/10.3389/fbioe.2019.00259.
FARAH, Shady; ANDERSON, Daniel G.; LANGER, Robert. Physical and mechanical properties of PLA, and their functions in widespread applications — A comprehensive review. Advanced Drug Delivery Reviews, [S.L.], v. 107, p. 367-392, Dez. 2016. Elsevier B.V. http://dx.doi.org/10.1016/j.addr.2016.06.012.
HANNA, J. R., & GIACOPELLI, J. A. (1997). A review of wound healing and wound dressing products. Journal of Foot and Ankle Surgery, 36(1), 2–14. https://doi.org/10.1016/S1067-2516(97)80003-8.
GASPARILLA, A.J.; MARTINEZ, G.A.; LUNELLI, B.H.; JARDINI, A.L.; FILHO, R.M. Poly-lactic acid synthesis for biomedical applications devices — A review. Biotechnol. Adv. 30 (2012) 321–328
MULLER, R., 2008. Biodegradability of polymers: Regulations and methods of testing. In: Steinbüchel, A. (Ed.), Biopolymers, General Aspects, and Special Applications, vol. 10., pp. 366–388 (Chapter 12).
- In what kind of direct utilization can be translated by these reported materials?
Answer: As a potential biologically rich biomembrane, it can be used mainly for skin wound healing.
- I expect the author to address precise biofunctional applications in the abstract and conclusion sections.
Answer: Thanks a lot for the comments. The first step is developing and characterizing this new membrane, loaded with the bioactive red propolis extract. The investigation is underway and will be published elsewhere. It is worth mentioning that it has potential application once it is stable and the release of the active principle is viable, as shown in the antioxidant activity assays. However, at the present moment, it is not possible to report biofunctional applications.
The summary and conclusion section described the potential applications of the developed membrane.
Reviewer 2 Report
Development of a Polymeric Membrane Impregnated with Poly-Lactic Acid (PLA) Nanoparticles Loaded with Red Propo-lis (MS No. 1935741).
The manuscript entitled “Development of a Polymeric Membrane Impregnated with Poly-Lactic Acid (PLA) Nanoparticles Loaded with Red Propo-lis” is a very promising and extraordinary work. The manuscript is designed well and performed nicely.
The manuscript could be accepted with following comments if incorporated in the revised manuscript.
1. The authors need to provide a comparative table for the performance of the proposed developed material with reported polymeric materials (NP) with and without Propo-lis. The authors need to discuss the merits of the proposed material compared with report materials.
2. (a) For application purposes, the authors need to apply the developed material and involved materials against bacteria and fungi along with control.
Optional
(b) The authors need to apply the developed material and involved materials on animal model (mice) to check the efficiency of developed material in terms of skin wound healing purpose if possible.
Author Response
The manuscript entitled "Development of a Polymeric Membrane Impregnated with Poly-Lactic Acid (PLA) Nanoparticles Loaded with Red Propolis" is an excellent and extraordinary work. The manuscript is designed well and performed nicely.
The manuscript could be accepted with the following comments if incorporated in the revised manuscript.
- The authors need to provide a comparative table for the performance of the proposed developed material with reported polymeric materials (NP) with and without propolis. The authors need to discuss the merits of the proposed material compared with report materials.
Answer: Excellent comment. See table 7 and the new text. The table was included in the text. Thanks a lot.
The literature currently reports different polymeric dressings to coat and accelerate tissue regeneration. The main ones are foam dressings, films, hydrogels, and hydrogel-based membranes developed with alginate, pectin, CMC, hydrocolloids, or other polymers such as PLA (Table 7) and have advantages and disadvantages inherent to the type of polymer matrix [32]. Hydrogel-based membranes arouse immense interest. Their high swelling capacity favors the absorption and retention of exudate, controlling the amount of fluids under the wound, maintaining adequate wettability and humidity, supporting the proliferation of fibroblasts, and the migration of keratinocytes [33].
Table 7. Polymeric wound dressings: Advantages and disadvantages.
|
Types of polymeric dressings |
Polymer matrix |
Characteristics |
Advantages |
Disadvantages |
Ref. |
|
Foam |
Polyurethane |
Flexible polyurethane foam and lignan nanoparticles enriched with green propolis |
Antibacterial activity due to propolis allows cell adhesion, is hydrophilic, and accelerates healing. |
Very adherent, forming an opaque layer that makes it difficult to monitor the wound, not applicable to dry wounds and with low stability. |
[34] |
|
Film |
Pectin |
Pectin film enriched with allantoin, plasticized with glycerol |
Absorbs exudate, flexible, proper contact angle, and accelerates healing |
Opaque layer formation, intermediate swelling ability |
[35] |
|
Hydrogel membrane |
Polyvinyl alcohol (PVA) |
PVA membrane enriched with red propolis |
Antibacterial activity due to propolis, high swelling, and traction capacity |
Variable mechanical and swelling stability, opaque, semipermeable to gases and water vapor |
[36] |
|
Sodium carboxymethylcellulose |
Sodium carboxymethylcellulose membrane enriched with red propolis |
Antibacterial activity due to propolis, high swelling capacity, antiallergic, low cost |
[4] |
||
|
Alginate membrane |
Alginate |
Alginate and PVA membrane enriched with curcumin nanoparticles |
Antibacterial activity due to curcumin, high swelling capacity, transparent, allows gas exchange |
Swelling stability and variable mechanics |
[37] |
|
Poly(lactic acid) membrane |
Poly(lactic acid) |
Polylactic acid membrane enriched with babassu oil |
Formed by pores, it maintains moisture on top of the wound, is non-toxic, promotes cell migration, and has antimicrobial activity. |
Low swelling capacity, formation of an opaque layer |
[38] |
|
Poly(lactic acid) (PLA)/ Sodium alginate (SA) / Poly(vinyl alcohol) (PVA) |
PLA/PVA/SA Fiber Membranes |
Mechanical stability, biocompatible and biodegradable, accelerates wound healing. |
[39] |
||
|
Poly(lactic acid) |
Polylactic acid membrane enriched with green propolis |
Mechanical stability, biocompatible and biodegradable, with antimicrobial activity |
[40] |
In the last decade, polymeric membranes have attracted a lot of attention due to their great potential for application in several areas, such as biomedical and pharmaceutical applications and the field of biotechnology. The literature reveals that membranes loaded with bioactive components have shown wound-healing potential. In addition, the association with polymeric, synthetic, biodegradable, and bioabsorbable nanomaterials such as PLA has shown promise for tissue recovery [5]. This study developed a topical membrane formulation based on NaCMC impregnated with PLA nanoparticles loaded with Brazilian red propolis. It was characterized in terms of its physicochemical and thermal aspects.
The membrane system developed here presents a differential concerning the different types described in the literature. We performed the incorporation of nanoparticles loaded with red propolis from Alagoas to improve the mechanical and thermal stabilities as a function of the swelling and vapor exchange capacity, as described in the literature.[32]. In addition, the membrane becomes functional by presenting the release of the nanoparticle and conferring a significant antioxidant activity based on the RPE. It is suggested that the biological properties of red propolis are also present in nanoparticles with RPE. There is no development in the literature on this type of membrane system (CMC + PLA nanoparticles + RPE). In the second stage of this project, mechanical and biological tests will be able to confirm these properties. Significantly, PLA nanoparticles, even in the absence of RPE, could accelerate the healing process. In a comparative study carried out by Bi et al. [39], using PLA, and PLA/PVA/SA electrospun fiber membranes for wound healing in vitro and in vivo, observed that isolated PLA fibers allowed to accelerate wound healing in rats. The developed membrane system will be able to act synergistically between the hydrophilic membrane (promoting exudate absorption) [32], PLA (protein deposition) [39], and RPE (antioxidant, anti-inflammatory, and healing activity) [4]. The analyses described in this work reveal that the developed membrane has characteristics suitable for its use in vitro and in vivo biological assays.
- (a) For application purposes, the authors need to apply the developed material and involved materials against bacteria and fungi along with control.
Answer: Excellent comment. We apologize. Our goals in this present study are to characterize the nanoparticle and membrane developed and to identify whether there would be interactions between the polymeric matrices and the chemical constituents of the RPE, allowing us to obtain a membrane with antioxidant activity. The investigation of this membrane's possible in vitro and in vivo application would require extra time, which is unfeasible at this stage.
Optional
(b) The authors need to apply the developed material and involved materials to an animal model (mice) to check the efficiency of advanced material in skin wound healing, if possible.
Thanks for the excellent suggestion. It will be our next step.
Round 2
Reviewer 1 Report
I recommend this paper for the publication